# Ancestral reconstruction of duplicated signaling proteins reveals the evolution of signaling specificity

Isabel Nocedal[1], Michael T Laub[1,2]*

[1]Department of Biology, Massachusetts Institute of Technology, Cambridge, United States; [2]Howard Hughes Medical Institute, Massachusetts Institute of Technology, Cambridge, United States

**Abstract** Gene duplication is crucial to generating novel signaling pathways during evolution. However, it remains unclear how the redundant proteins produced by gene duplication ultimately acquire new interaction specificities to establish insulated paralogous signaling pathways. Here, we used ancestral sequence reconstruction to resurrect and characterize a bacterial two-component signaling system that duplicated in α-proteobacteria. We determined the interaction specificities of the signaling proteins that existed before and immediately after this duplication event and then identified key mutations responsible for establishing specificity in the two systems. Just three mutations, in only two of the four interacting proteins, were sufficient to establish specificity of the extant systems. Some of these mutations weakened interactions between paralogous systems to limit crosstalk. However, others strengthened interactions within a system, indicating that the ancestral interaction, although functional, had the potential to be strengthened. Our work suggests that protein-protein interactions with such latent potential may be highly amenable to duplication and divergence.

*For correspondence:
laub@mit.edu

## Editor's evaluation

It is thought that gene duplications are a major factor driving the emergence and expansion of paralogous proteins, but how redundant proteins evolve to acquire new interaction specificities is poorly understood. Here the authors focused on understanding how specificity evolves in signaling pathways involving bacterial two-component systems. They identify three residues involved in preventing cross-talk between non-cognate pairs of two pairs of paralogous proteins. Remarkably, they show that changes in these 3 residues were sufficient for the establishment of specificity and establishment of insulation of two signaling pathways following a duplication event.

## Introduction

Protein-protein interactions are critical for most cellular functions, including signal transduction pathways. Notably, many protein interaction domains in signaling proteins are members of large paralogous families. For example, in mammals there are dozens of SH2, SH3, and PDZ domains that each mediate a variety of protein-protein interactions (*Pawson, 2004*). These paralogous proteins arise through a process of gene or domain duplication, which enables organisms, on an evolutionary timescale, to rapidly expand their signaling repertoires (*Alm et al., 2006*; *Corrochano et al., 2016*). However, the use of paralogous proteins and domains comes at a cost, requiring cells to avoid deleterious crosstalk between highly similar proteins and domains (*Bradley and Beltrao, 2019*; *Capra et al., 2012*; *Siryaporn and Goulian, 2008*; *Zarrinpar et al., 2003*). How the process of duplication

and divergence unfolds at a molecular level to ensure the specificity of paralogous signaling proteins is not clear.

The precise mutations required to produce highly specific protein-protein interactions upon duplication is largely unexplored, in part because the underlying duplication events that produced most extant paralogs were ancient events. One approach to tackling this problem involves ancestral protein reconstruction, which uses the phylogenies of extant proteins to infer the sequences of ancient proteins (*Hochberg and Thornton, 2017*). This approach has been powerfully applied to examine the evolution of protein-ligand interactions (*Voordeckers et al., 2012*), such as steroid hormones and their receptors (*Bridgham et al., 2009*; *Bridgham et al., 2006*), transcription factor-DNA interactions (*Baker et al., 2013*; *McKeown et al., 2014*; *Starr et al., 2017*), and protein-drug interactions (*Wilson et al., 2015*). There have been fewer studies applying ancestral protein reconstruction to protein-protein interactions (*Holinski et al., 2017*; *Laursen et al., 2021*; *Wheeler et al., 2018*; *Wheeler and Harms, 2021*), with most focusing on resurrecting the mutations that impact protein oligomerization (*Hochberg et al., 2020*; *Pillai et al., 2020*).

We sought to understand how paralogous protein-protein interactions arise through duplication and divergence, focusing on bacterial two-component signaling pathways. Two-component signaling systems typically consist of a sensor histidine kinase (HK) that autophosphorylates upon signal recognition, and then transfers a phosphoryl group to a cognate response regulator (RR) that can trigger an intracellular response, frequently through changes in gene expression (*Figure 1A*; *Buschiazzo and Trajtenberg, 2019*; *Capra and Laub, 2012*). Most bacteria encode dozens of these systems, with each system usually insulated from every other paralogous system (*Galperin, 2005*; *Koretke et al., 2000*; *Skerker et al., 2005*). Crosstalk between systems appears relatively rare, and has been shown to produce fitness defects when introduced artificially (*Capra et al., 2012*), likely due to the decreased ability of a given HK to activate a given RR (*Rowland and Deeds, 2014*). The specificity of the HK-RR interaction is determined primarily through molecular recognition, with a relatively small number of amino acids in both the histidine kinase and response regulator promoting the cognate interaction and preventing unwanted crosstalk with non-cognate proteins (*Capra et al., 2012*; *Skerker et al., 2008*).

Despite their prevalence in bacterial genomes and the prior identification of the key specificity-determining residues, it remains unclear how crosstalk between recently duplicated HK-RR pairs is eliminated to establish two insulated pathways. While previous work demonstrated how extant proteins can be rewired to recognize different substrates (*Capra et al., 2010*; *Skerker et al., 2008*), it is unclear whether duplicated systems resolve crosstalk in a similar way. In particular, it is not known how many of the four proteins involved must acquire mutations to insulate the two protein-protein interfaces. In principle, there are two general models for how interacting proteins could evolve specificity after a duplication event (note that HK-RR systems are typically co-operonic and likely duplicate as an operon) (*Figure 1B*). In the first model, mutations could occur in both proteins of one system to retain their compatibility with each other while rendering them incompatible with proteins in the other system, leading to insulated pathways. This model is akin to neofunctionalization, in which a new, unique interface evolves in one paralogous system. In the second model, changes in all four proteins, perhaps each of smaller magnitude, could be required to insulate these systems. This model could represent neofunctionalization of both systems or subfunctionalization, where the functions of an ancestral protein are divided between paralogs.

Here, we use ancestral protein reconstruction to resurrect extinct HK and RR proteins that existed prior to a two-component system duplication event. By characterizing these ancestors, as well as mutational intermediates that descended from them, we elucidate the likely mutational trajectories taken by these proteins that resulted in the insulation of their protein-protein interfaces. We find that just three mutations can largely account for the establishment of specificity in the two pathways. Unexpectedly, these three mutations occur in the HK of one system and the RR of the other. The mutations that arise in one of the HKs serve mainly to prevent crosstalk to the RR of the other system whereas the mutation that arises in the RR serves both to prevent crosstalk and promote interaction with its cognate HK, suggesting that the HK-RR system that existed pre-duplication had the potential for faster phosphotransfer. By exploiting this latent ability to improve the interaction, along with the emergence of mutations that directly block crosstalk, these two HK-RR systems evolved specificity. Thus, our results reveal the likely mutational

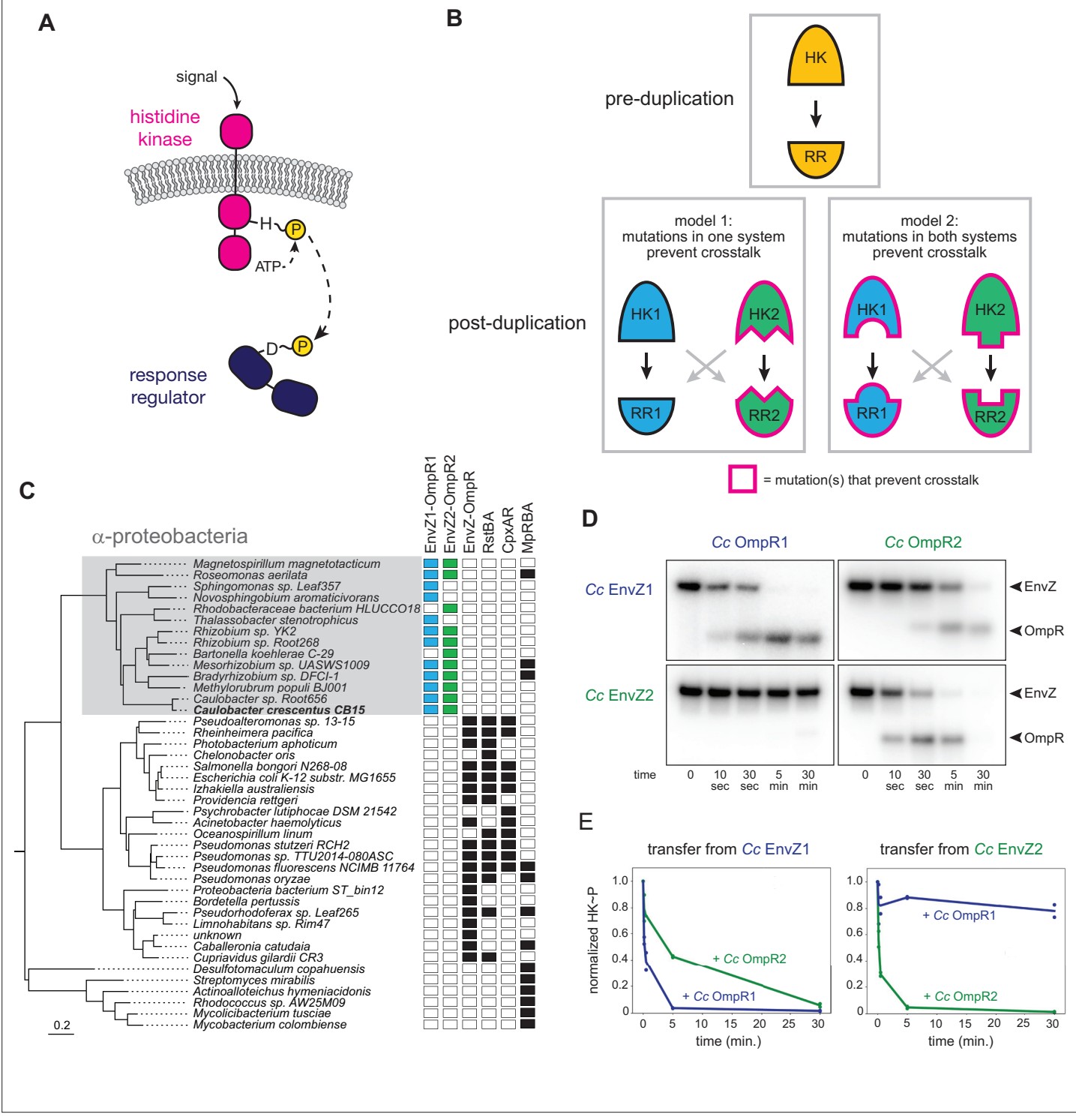

**Figure 1.** EnvZ-OmpR duplication and divergence in *C. crescentus*. (**A**) Two-component signal transduction. A sensor histidine kinase autophosphorylates upon activation and transfers a phosphoryl group to a cognate response regulator to activate an intracellular response. (**B**) Two models for acquisition of paralog specificity after duplication of an interacting histidine kinase (HK) and response regulator (RR). Model 1: both proteins in one system acquire compensatory mutations that maintain their interaction while preventing interaction with the other system. Model 2: all four proteins acquire mutations that prevent crosstalk between systems. (**C**) Phylogenetic species tree of proteobacteria inferred from 27 ribosomal protein sequences showing distribution of EnvZ-OmpR homologs and related systems. Scale bar indicates substitutions per site. (**D**) In vitro phosphotransfer specificity of *C. crescentus* EnvZ and OmpR paralogs. Purified histidine kinase (cytoplasmic domain only) was autophosphorylated and then mixed with a given purified response regulator and incubated for the time indicated. At t=0 a single upper band corresponds to the autophosphorylated HK. At

*Figure 1 continued on next page*

*Figure 1 continued*

subsequent time points, a second, lower band corresponding to the RR appears as the kinase transfers its phosphoryl group leading to depletion of the autophosphorylated HK. At longer time points phosphatase activity of the HK can lead to disappearance of the phosphorylated RR. (**E**) Quantification of phosphorylated HK over time in (**D**). Values were normalized to t=0 for each HK-RR pair. Lines represent mean (n=2) and dots represent independent replicates.

The online version of this article includes the following source data and figure supplement(s) for figure 1:

**Source data 1.** Quantified phosphotransfer values for *Figure 1E*.

**Figure supplement 1.** Comparison of extant EnvZ-OmpR paralogs.

**Figure supplement 2.** Phosphorylation of histidine kinases alone.

trajectory responsible for the rapid establishment of specificity in paralogous proteins immediately post-duplication.

## Results

## EnvZ/OmpR has undergone duplication and diversification in α-proteobacteria

EnvZ-OmpR is a widespread two-component signaling system that has been best characterized in *E. coli* (*Cai and Inouye, 2002*). Many, though not all, α-proteobacteria contain two paralogous EnvZ-OmpR systems that appear to descend from a duplication that occurred in a basal α-proteobacterium (*Figure 1C*). In *Caulobacter crescentus* these two systems are CC1181-1182 and CC2932-2931, which we hereafter refer to as EnvZ1-OmpR1 and EnvZ2-OmpR2. These systems share less than 50% sequence identity, with particularly high divergence in the sensory domains (*Figure 1—figure supplement 1A*). The duplication of EnvZ-OmpR in α-proteobacteria correlates with an absence of the related two-component signaling systems RstAB and CpxAR, which are highly conserved in the γ-proteobacteria (*Figure 1C*).

To determine the specificity of the EnvZ1-OmpR1 and EnvZ2-OmpR2 systems at the level of phosphotransfer, we used biochemical assays with purified proteins in vitro. For EnvZ1 and EnvZ2, we purified the cytoplasmic, catalytic domains (DHp and CA) of each kinase fused to an N-terminal MBP-His$_6$ tag. For OmpR1 and OmpR2, we purified each full-length response regulator harboring an N-terminal Trx-His$_6$ domain. To assess phosphotransfer, a given histidine kinase was first incubated with [γ-$^{32}$P]-ATP to drive autophosphorylation and then mixed at a 1:4 molar ratio with a response regulator of interest. Samples taken at various time points were examined by SDS-PAGE. At t=0, before adding a response regulator, there was a single band corresponding to the autophosphorylated kinase. At subsequent time points, a second band appeared as the kinase transferred its phosphoryl group to the response regulator; efficient transfer eventually led to depletion of the autophosphorylated kinase. Histidine kinases are typically bifunctional, such that when not autophosphorylated they can drive the dephosphorylation of a response regulator. This activity usually occurs on a slower timescale than phosphotransfer, and explains why the bands corresponding to phosphorylated response regulator often decreased at later time points in our assays (*Figure 1D*).

Using these in vitro phosphotransfer assays, we first tested the specificity of each paralogous system from *C. crescentus*. EnvZ1 transferred rapidly to OmpR1, its cognate response regulator, with phosphorylated OmpR1 detected after 10 s and autophosphorylated EnvZ depleted by 5 min (*Figure 1D*, top left). Autophosphorylated EnvZ1 also transferred to OmpR2, but less efficiently as indicated by the slower accumulation of phosphorylated OmpR2 and slower depletion of autophosphorylated EnvZ1, with nearly full depletion occurring only by 30 min (*Figure 1D*, top right). Similar patterns were observed for EnvZ2, which transferred very rapidly to OmpR2, but quite slowly to OmpR1 (*Figure 1D*).

To compare the rates of transfer from different histidine kinases, we quantified the level of autophosphorylated kinase in each phosphotransfer reaction over time (*Figure 1E*). The rate at which the autophosphorylated kinase decreases is a proxy for the rate of phosphotransfer. (Each histidine kinase is stably phosphorylated over 30 min (*Figure 1—figure supplement 2*)). By this measure, each EnvZ paralog exhibited a clear preference for its cognate response regulator (*Figure 1E*). Although EnvZ1 shows a weaker preference for its cognate response regulator in these assays than does EnvZ2, EnvZ1

showed a strong preference for OmpR1 in an in vitro competition assay (*Figure 1—figure supplement 1B*), and even modest substrate preferences in vitro can result in significant in vivo insulation (*Capra et al., 2012*; *McClune et al., 2019*). We conclude that each histidine kinase has a preference for its cognate, co-operonic response regulator compared to the paralogous response regulator. Further, these results indicated that since the duplication event that created the EnvZ-OmpR paralogs in α-proteobacteria, each protein-protein interaction has diverged to generate paralog specificity.

## Ancestral protein reconstruction reveals early acquisition of paralog specificity

To determine the evolutionary trajectory that resulted in the diversification and phosphotransfer insulation of the EnvZ-OmpR paralogs in α-proteobacteria, we used ancestral protein reconstruction to infer the sequences of the ancestral proteins. A maximum likelihood phylogeny was inferred for 200 matched pairs of cognate histidine kinase-response regulators from the EnvZ-OmpR family and other closely related two-component signaling system families (*Figure 2A*; full phylogeny in *Figure 2—figure supplement 1*). This matched HK-RR phylogeny was found to be largely concordant with phylogenies based on HK or RR sequences alone (*Figure 2—figure supplement 2*). Based on the matched HK-RR phylogeny, we identified the maximum a posteriori EnvZ (catalytic domains only) and OmpR sequences immediately prior to the duplication, and immediately after the duplication (*Figure 2B–C*; full alignments in *Figure 2—figure supplement 3A–B*). This duplication event was quite ancient, having occurred near the origin of the α-proteobacteria ~1900 million years (Ma) ago (*Wang and Luo, 2021*), and these ancestral sequences share only ~50% identity with the extant proteins in *C. crescentus* (*Figure 3—figure supplement 2A–B*). The last common ancestral proteins from which both EnvZ-OmpR paralogs descend will be referred to as ancHK and ancRR, while ancHK1-ancRR1 refers to the ancestor of all EnvZ1-OmpR1 proteins and ancHK2-ancRR2 is the ancestor of all EnvZ2-OmpR2 (*Figure 2A*). Each ancestor was reconstructed with high confidence, with mean posterior probabilities >0.8 (ancHK=0.85, ancRR=0.88, ancHK1=0.87, ancRR1=0.91, ancHK2=0.81, ancRR2=0.85; *Figure 3—figure supplement 2C*). Furthermore, the sequences generated from the matched phylogeny were highly similar to those generated from an ancestral reconstruction of HK or RR sequences individually (*Figure 2—figure supplements 4 and 5*). Each of the reconstructed ancestral sequences was cloned, expressed, and then purified, as above.

We first tested our ancestral proteins for activity in our in vitro phosphotransfer experiments, and found that all show clear activity in this assay, indicating that the inferred ancestors represent functional histidine kinases and response regulators. Importantly, we observed transfer from ancHK to ancRR (*Figure 2D*), indicating that even for our most ancient reconstructions we generated proteins capable of interacting and engaging in a productive phosphotransfer event.

We next sought to determine whether specificity, at the level of phosphotransfer, had emerged immediately after the duplication event in ancHK1-ancRR1 and ancHK2-ancRR2, systems that share ~70% identity (*Figure 3—figure supplement 2B*). To do so, we measured phosphotransfer in vitro from autophosphorylated ancHK1 and ancHK2 to ancRR1 and ancRR2 (*Figure 3A*). We found that each ancestral kinase robustly phosphorylated its reconstructed cognate partner, with complete transfer for ancHK1-ancRR1 after 5 min and almost complete transfer for ancHK2-ancRR2 after 5 min. In contrast, each kinase showed slower transfer to the non-cognate regulator. Similar to their orthologous counterparts in *C. crescentus*, ancHK2 showed a stronger cognate preference, with very little transfer to ancRR1 observed at 5 min. While ancHK1 transfers more rapidly to its non-cognate regulator than ancHK2 transfers to its non-cognate regulator, a clear preference was still observed after 5 min, with autophosphorylated ancHK1 fully depleted after mixing with ancRR1 but not fully depleted when mixed with ancRR2. These results indicated that phosphotransfer specificity was established in ancHK-ancRR1 and ancHK-ancRR2 shortly after the duplication of ancHK-ancRR.

To determine which of the proteins acquired mutations that prevented crosstalk between paralogous systems, we first examined the phosphotransfer properties of ancHK and ancRR. We found that ancHK transferred robustly to ancRR (*Figure 2D*), as well as to ancRR1 and ancRR2 (*Figure 3B*), and that ancHK1 and ancHK2 both transferred robustly to ancRR (*Figure 3D*). We then compared the ability of ancestral histidine kinases pre- and post-duplication to transfer to a given post-duplication response regulator. We found that transfer from either ancHK or ancHK1 to ancRR2 was similar (*Figure 3C*, right), suggesting that ancHK1 did not acquire mutations to prevent crosstalk with

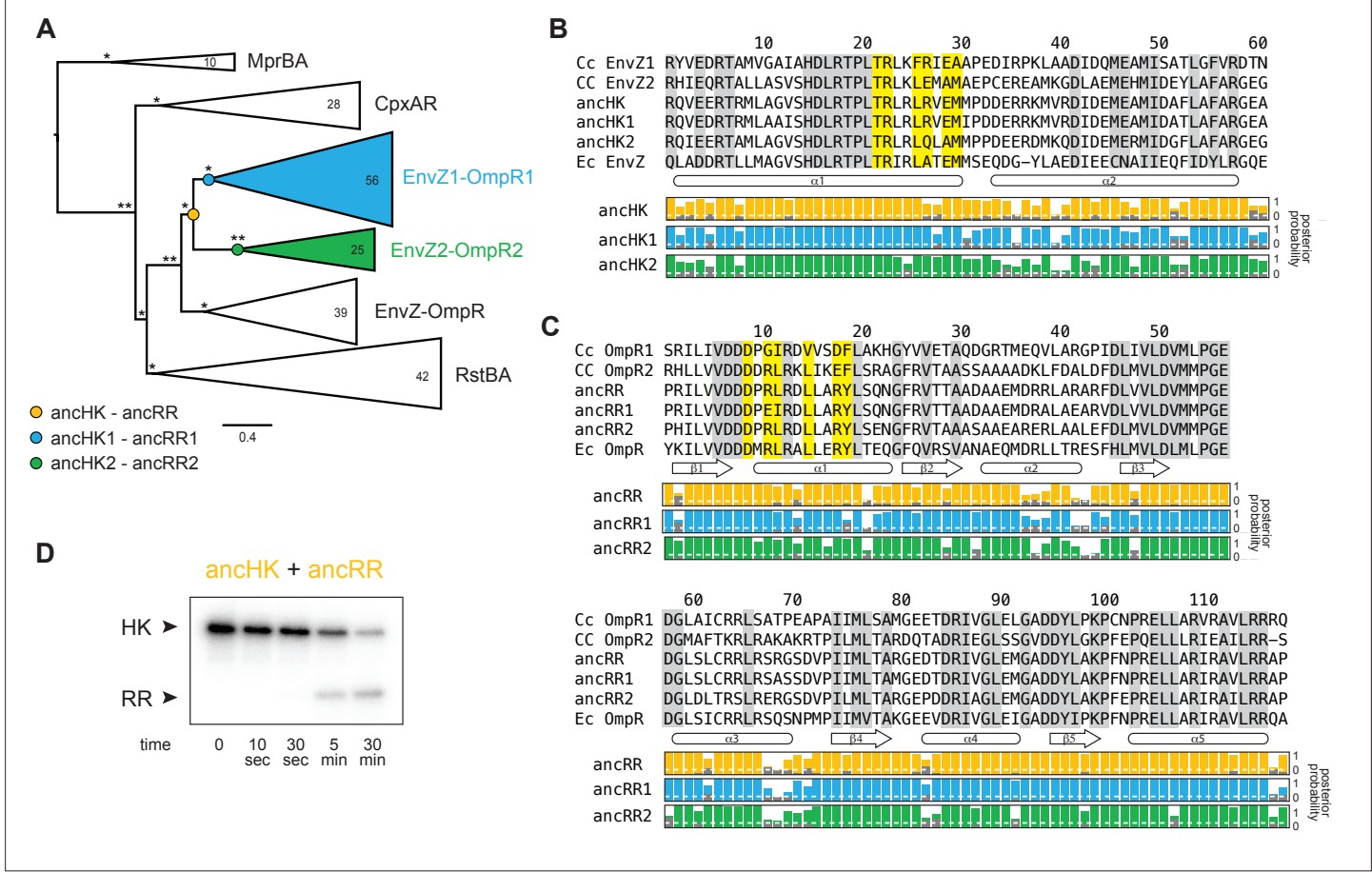

**Figure 2.** Inference of ancestral α-proteobacterial EnvZ-OmpR proteins. (**A**) Simplified phylogenetic tree of merged, matched EnvZ and OmpR sequences. Number of sequences in each clade indicated. Node support indicated by approximate likelihood ratio statistic (* indicates >10, ** indicates >100). Circles represent reconstructed ancestral protein pairs. Scale bar represents substitutions per site. For complete phylogeny, see *Figure 2—figure supplement* 1. (**B–C**) Multiple sequence alignment of EnvZ DHp domains (**B**) and OmpR receiver domains (**C**) from extant *C. crescentus* paralogs, ancHK-ancRR, ancHK1-ancRR1, ancHK2-ancRR2, and extant *E. coli* EnvZ-OmpR sequences. Residues conserved in both *C. crescentus* paralogs and all ancestors highlighted in grey; residues previously shown to strongly covary and dicatate specificity in *E. coli* EnvZ (*Capra et al., 2010*) highlighted in yellow. Secondary structure elements, based on AlphaFold prediction of the ancHK-ancRR complex shown below alignment. Posterior probabilities of reconstructed ancestral sequences at these positions shown for ancHK-RR (yellow), ancHK1-RR1 (blue), and ancHK2-RR2 (green) with most likely residue indicated by respective colors, and second most likely shown in grey. Dashed white line indicates posterior probability of 0.2, the threshold for identifying sites to be alternatively reconstructed (see *Figure 2—figure supplement 4* and 5). (**D**) Phosphotransfer from autophosphorylated ancHK to ancRR.

The online version of this article includes the following figure supplement(s) for figure 2:

**Figure supplement 1.** Phylogeny of EnvZ-OmpR paralogs and related systems.

**Figure supplement 2.** HK-only and RR-only phylogenies.

**Figure supplement 3.** Multiple sequence alignments of EnvZ and OmpR proteins.

**Figure supplement 4.** Alignment of ancestral HK sequences reconstructed using alternative techniques.

**Figure supplement 5.** Alignment of ancestral RR sequences reconstructed using alternative techniques.

ancRR2. In contrast, we found that ancHK2 transferred to ancRR1 much more slowly than did ancHK (*Figure 3C*, left), suggesting that ancHK2 must have acquired mutations post-duplication that prevent crosstalk with ancRR1 (all transfers shown together in *Figure 3—figure supplement 1*).

We then compared the ability of ancRR, ancRR1, and ancRR2 to be phosphorylated by ancHK1, finding that ancRR was not phosphorylated more rapidly than ancRR2 (*Figure 3E*, left), indicating that ancRR2 did not acquire mutations that prevent crosstalk with ancHK1. We also compared the ability of ancRR, ancRR1, and ancRR2 to be phosphorylated by ancHK2 (*Figure 3E*, right). In this case, we

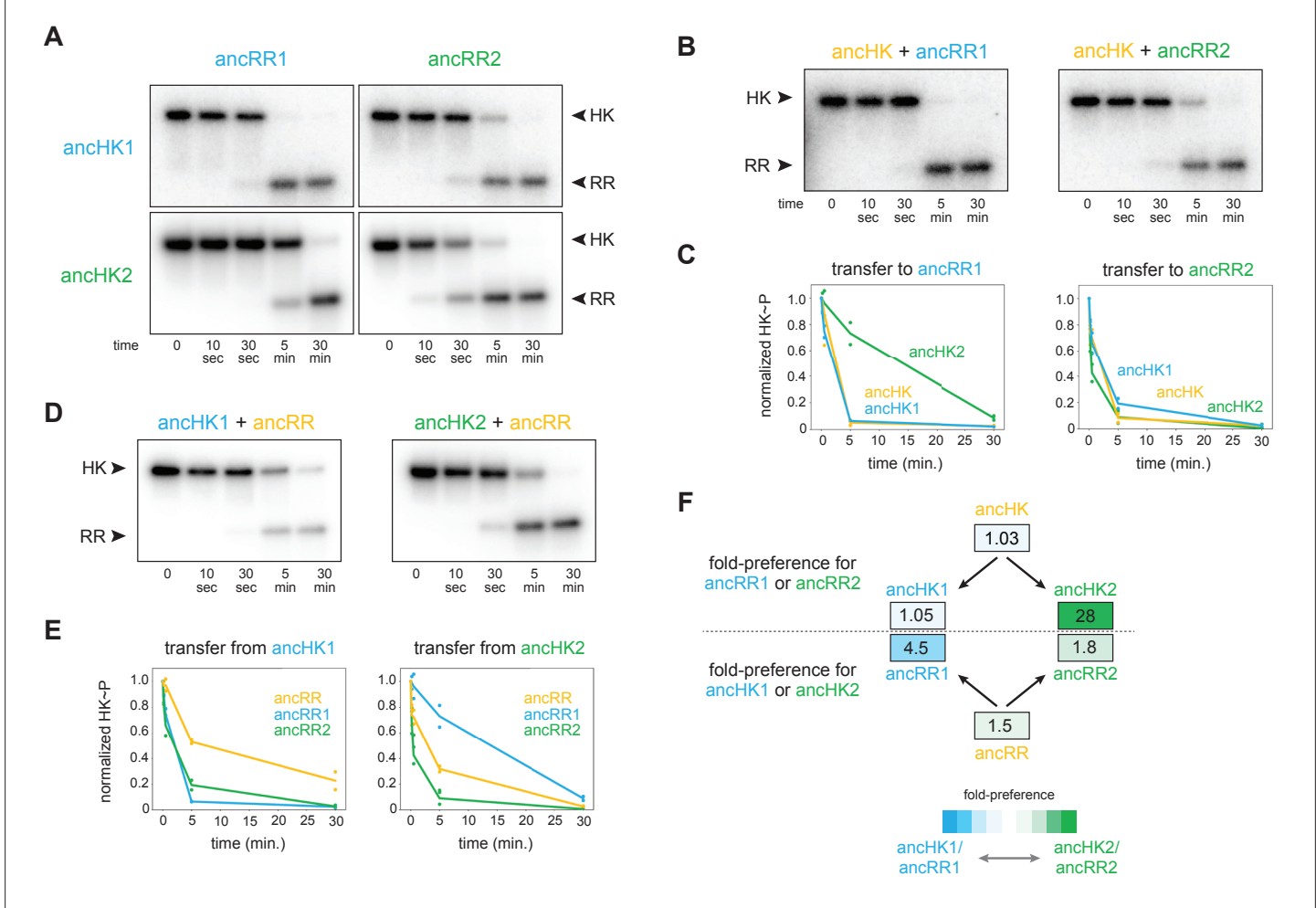

**Figure 3.** AncHK2 and ancRR1 acquired new specificities post-duplication. (**A**) Phosphotransfer from ancHK1 and ancHK2 to ancRR1 and ancRR2. (**B**) Phosphotransfer from ancHK to ancRR1 and ancRR2. (**C**) Quantification of the phosphorylated HKs indicated over time for (**A**) and (**B**) for transfer to ancRR1 (left) and ancRR2 (right). Lines represent mean (n=2) and dots represent independent replicates. (**D**) Phosphotransfer from ancHK1 and ancHK2 to ancRR. (**E**) Quantification of phosphorylated HK over time from (**A**) and (**D**) for transfer from ancHK1 (left) and ancHK2 (right) to the RRs indicated. (**F**) Estimate of substrate specificity for all ancestors. The ratio of specificity constants ($k_{cat}/k_M$) was determined for each HK or RR using the initial rate of phosphotransfer to one protein relative to another. Blue indicates a preference for ancHK1 (for RRs) or ancRR1 (for HKs), green represents a preference for ancHK2 (for RRs) or ancRR2 (for HKs), and white indicates no preference. Numbers indicate fold-preference (ratio of specificity constants).

The online version of this article includes the following source data and figure supplement(s) for figure 3:

**Source data 1.** Quantified phosphotransfer values for *Figure 3C*.

**Source data 2.** Quantified phosphotransfer values for *Figure 3E*.

**Source data 3.** Quantified phosphotransfer values for *Figure 3—figure supplement 2E*.

**Figure supplement 1.** All Phosphotransfer Reactions Compared.

**Figure supplement 2.** Ancestral protein reconstruction details.

find that ancRR1 was phosphorylated much more slowly than ancRR, indicating that ancRR1 must have acquired mutations that prevent crosstalk with ancHK2. These findings were robust to phylogenetic uncertainty, as we observed a similar pattern with the alternative ancestors (*Figure 3—figure supplement 2D-E*). We concluded that of the two response regulators produced by a duplication of ancRR only ancRR1 acquired mutations promoting insulation of the two paralogous pathways.

To quantify the change in specificity among ancestral proteins, we measured initial rates of phosphotransfer to estimate ratios of specificity constants. For a given histidine kinase, the ratio of specificity constants ($k_{cat}/k_M$) for two response regulators represents an approximate measure of substrate

specificity, and likewise for a given response regulator the ratio of transfer from two different histidine kinases represents an approximate measure of specificity. Comparing the ratios of specificity constants of the ancestral histidine kinases (*Figure 3F*), we found that both ancHK and ancHK1 showed little kinetic preference, whereas ancHK2 showed an ~28-fold kinetic preference for ancRR2 relative to ancRR1, supporting the idea that the kinetic preference of ancHK1 did not change significantly post-duplication while that of ancHK2 did. For the response regulators, we observed the opposite pattern, with both ancRR and ancRR2 showing modest kinetic preferences for transfer from ancHK2 (~1.5 and 1.8-fold, respectively) and ancRR1 showing a strong preference for ancHK1 (~4.5 fold) (*Figure 3F*). Taken all together, our results indicate that just two of the four paralogs, ancHK2 and ancRR1, acquired mutations that significantly alter their protein-protein interaction specificity in order to prevent cross-talk between the paralogous systems.

Reconstructing ancestral proteins is inherently probabilistic, and there is a degree of uncertainty associated with any reconstructed protein. To ensure that our conclusions were robust to this uncertainty, we reconstructed 'AltAll' alternative sequences for the six pre- and post-duplication ancestors using a previously described method (*Eick et al., 2017*). In short, for every position at which multiple residues had posterior probabilities >20%, the second most likely residue was included. These alternative ancestors were then tested for their ability to transfer to each other (*Figure 3—figure supplement 2D*). Some of the alternative ancestors transferred more slowly than the primary ancestors. However, as with the primary ancestors, we found that just two of the alternative ancestors, ancHK2-alt and ancRR1-alt, showed significantly different transfer specificity when compared to the pre-duplication ancestors (*Figure 3—figure supplement 2E–F*). This finding supports our conclusion that mutations in just two of the four paralogs were responsible for the insulation of these pathways.

## A small set of mutations was sufficient to insulate ancestral paralogs

To identify the individual mutations responsible for the change in specificity of ancHK2, we compared the sequence of ancHK to that of ancHK1 and ancHK2, focusing on six positions previously shown to strongly covary between histidine kinases and response regulators and to dictate the specificity of *E. coli* EnvZ-OmpR (*Capra et al., 2010*). Only two of these positions differ between ancHK and ancHK2: positions 27 and 29, which have changed from an arginine and glutamate to a glutamine and alanine, respectively (*Figure 4A*). To determine if these mutations are paralog-specific, and thus likely to be important in insulating these systems, we compared the amino acids at these two positions in all identified extant EnvZ1 and EnvZ2 orthologs (a much larger set of sequences than was used for our ancestral reconstructions). This analysis indicated that both positions are indeed strongly paralog specific. At position 27, arginine is present in >90% of 1,886 EnvZ1 sequences but in none of the 822 EnvZ2 sequences, for which >90% of sequences feature either glutamine, glutamate, or serine (*Figure 4B*). At position 29, the negatively charged residues glutamate and aspartate are present in >90% of EnvZ1 sequences but <10% of EnvZ2 sequences, where alanine is present in >85% of sequences (*Figure 4B*).

A similar analysis was performed to identify possible causal mutations in the evolution of ancRR1 specificity. Only two of the key positions were found to differ between ancRR and ancRR1 (*Figure 4C*), and only one of these, position 11, showed broad paralog specificity (*Figure 4D*). At this position, arginine is present in >90% of OmpR2 sequences and <1% of OmpR1 sequences. Instead, negatively charged glutamate and aspartate are present in >60% of OmpR1 sequences (*Figure 4D*). Importantly, all three of these potential key residues (27 and 29 in the kinase and 11 in the response regulator) were well supported positions in the reconstructed ancestors (*Figure 2B–C*), with none of them meeting the criteria for alternative reconstruction in the 'AltAll' alternative ancestors.

To determine if these positions were responsible for the insulation of the paralogous systems, we tested the effect of substitutions at these positions in ancHK and ancRR. We first introduced the substitutions R27Q and E29A separately and together into ancHK and measured phosphotransfer to ancRR1 (*Figure 4E*). Relative to the parental protein, ancHK, both individual substitutions slowed transfer to ancRR1, with significantly less transfer observed at 2 min. When combined, these two substitutions decreased transfer to ancRR1 further, with a rate of transfer now comparable to that observed with ancHK2 (*Figure 4E, H*). These substitutions did not have a significant effect on transfer to ancRR2 (*Figure 4—figure supplement 1A, B*). Thus, these two substitutions alone are sufficient to slow transfer from ancHK to ancRR1 and likely account for the major changes in ancHK2 that occurred post-duplication to help drive the insulation of the two paralogous pathways.

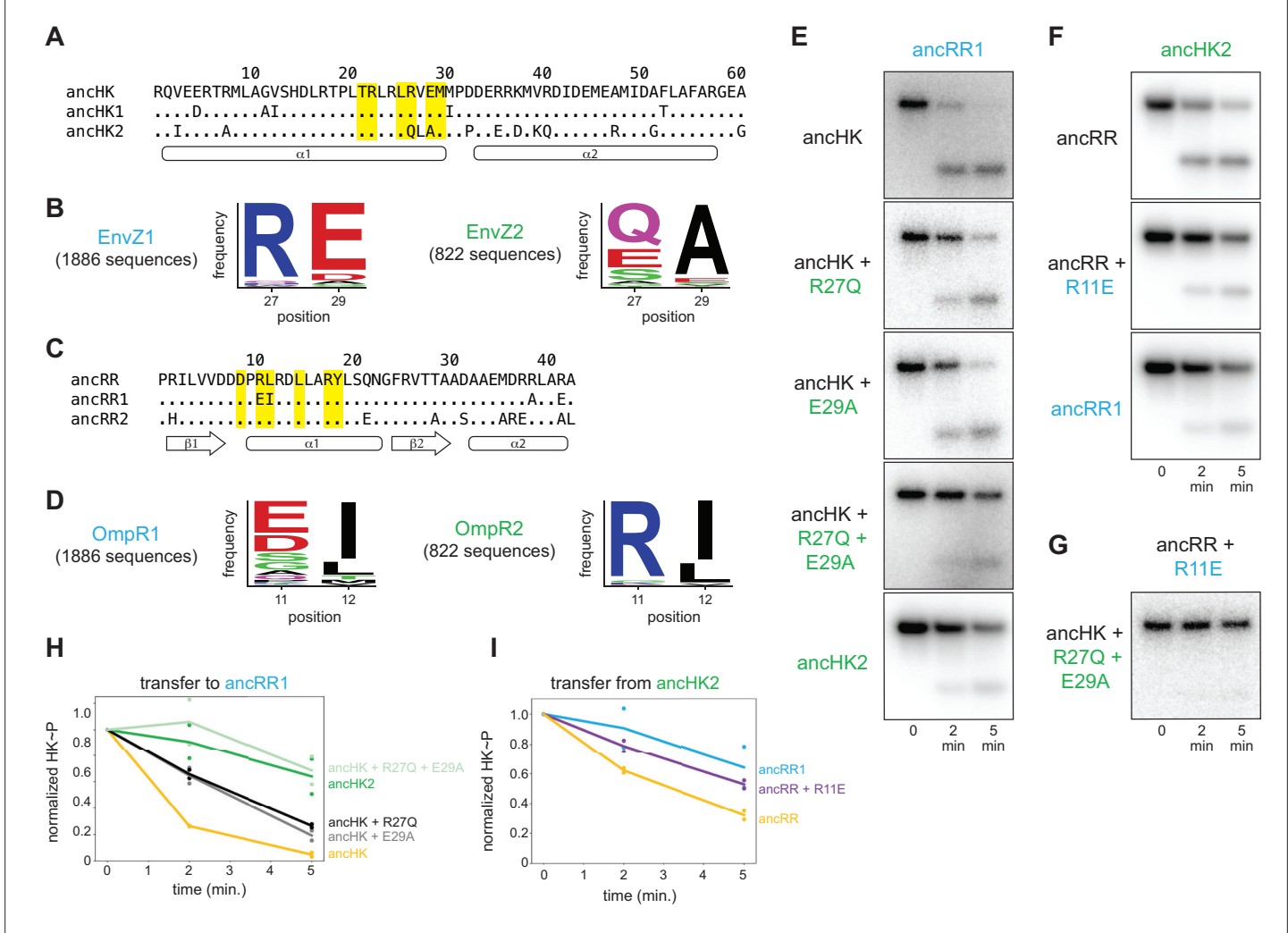

**Figure 4.** Identification of mutations responsible for ancestral paralog insulation. (**A**) Sequences of ancHK, ancHK1, and ancHK2 for regions primarily involved in molecular recognition. Dots indicate conservation compared to ancHK. Residues previously shown to be strongly coevolving and important for specificity of *E. coli* EnvZ-OmpR highlighted in yellow, and secondary structure elements predicted by AlphaFold indicated below sequence. (**B**) Sequence logos for HK positions 27 and 29 in 1,886 identified EnvZ1 paralogs and 822 identified EnvZ2 paralogs, with height indicating frequency of each amino acid. (**C**) Same as (**A**) but for ancRR, ancRR1, and ancRR2. (**D**) Same as (**B**) but for positions 11 and 12 in OmpR. (**E**) Phosphotransfer from ancHK, ancHK with the mutations indicated, and ancHK2 to ancRR1 at 0, 2 and 5 minute timepoints. (**F**). Phosphotransfer from ancHK2 to ancRR, ancRR + R11E, and ancRR1. (**G**). Phosphotransfer from ancHK + R27Q + E29A to ancRR + R11E. (**H–I**) Quantification of normalized phosphorylated HK from (**E**) and (**F**). Lines represent mean (n=2) and dots represent independent replicates.

The online version of this article includes the following source data and figure supplement(s) for figure 4:

**Source data 1.** Quantified phosphotransfer values for *Figure 4H*.

**Source data 2.** Quantified phosphotransfer values for *Figure 4I*.

**Source data 3.** Quantified phosphotransfer values for *Figure 4—figure supplement 1E*.

**Figure supplement 1.** Phosphotransfer analysis of mutations impacting paralog specificity.

We then tested the effect of the substitution R11E in ancRR on transfer from ancHK2 (*Figure 4F*). We found that introducing this single substitution into ancRR was sufficient to significantly slow transfer from ancHK2, with a rate of transfer very similar to that seen for ancHK2 to ancRR1 (*Figure 4F, I*). Finally, we examined all three of these substitutions together by testing transfer from ancHK(R27Q, E29A) to ancRR(R11E), finding very slow transfer (*Figure 4G*), as seen with ancHK2 and ancRR1. Together, these results demonstrate that just three mutations – two in ancHK and one in ancRR – are sufficient to confer specificity to these EnvZ-OmpR paralogs at the level of phosphotransfer. Further,

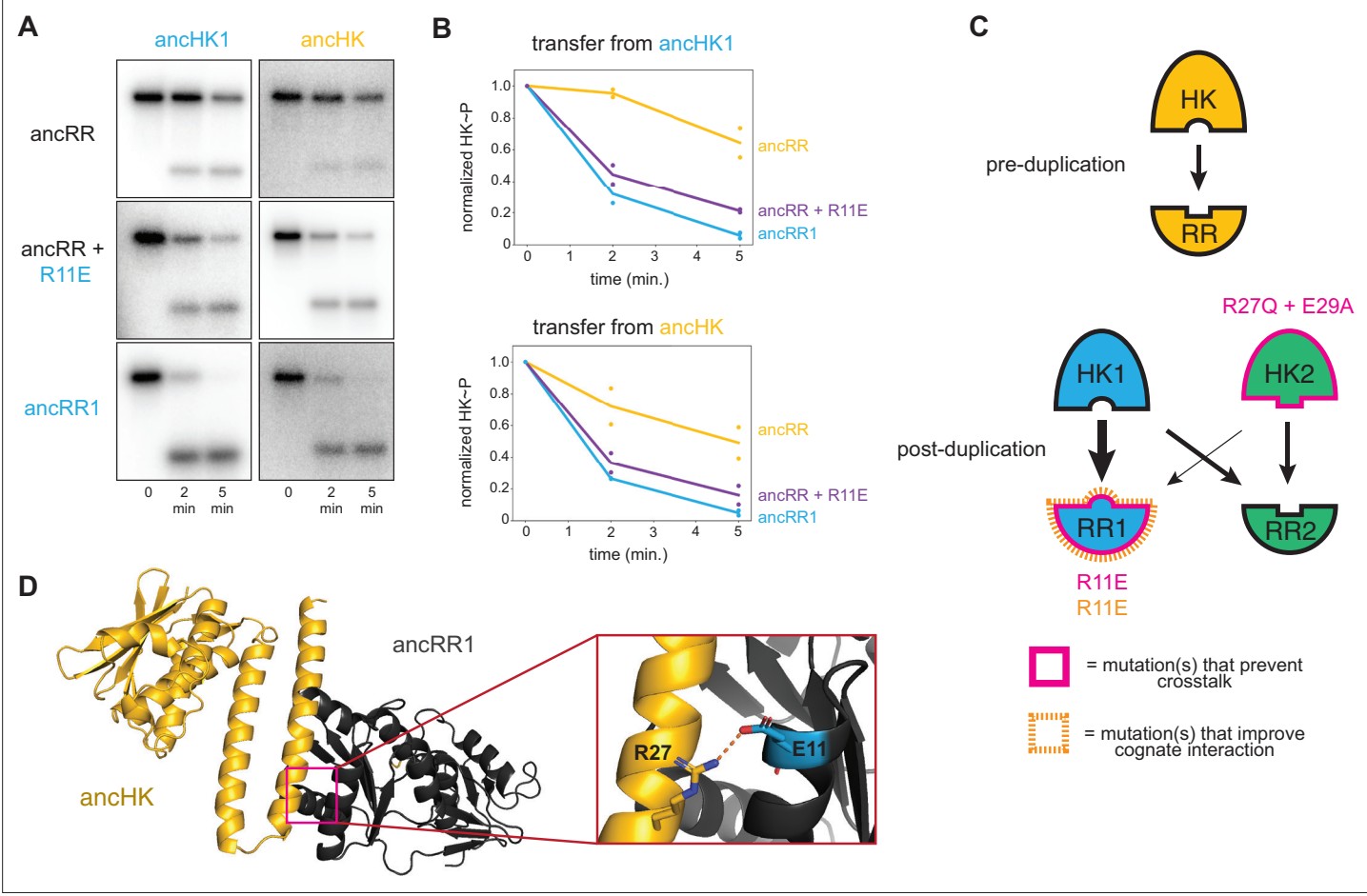

**Figure 5.** Identification of a mutation that enhances EnvZ-OmpR interactions. (**A**) Phosphotransfer from ancHK1 (left) and ancHK (right) to ancRR, ancRR + R11E, and ancRR1 at 0, 2, and 5-min timepoints. (**B**) Quantification of normalized phosphorylated HK from (**A**). (**C**) Model for insulation of EnvZ-OmpR paralogs in α-proteobacteria. Thickness of the black arrows indicates relative strength of a given interaction. Mutations that prevent crosstalk between paralogs indicated in pink; mutations that improve cognate interaction indicated in orange. (**D**) Predicted ancHK-ancRR1 complex structure from AlphaFold2. Inset: putative salt bridge between arginine 27 in ancHK and glutamate 11 in ancRR1 indicated by dashed line.

The online version of this article includes the following source data and figure supplement(s) for figure 5:

**Source data 1.** Quantified phosphotransfer values for *Figure 5B*.

**Source data 2.** PDB file for AlphaFold predicted ancHK-ancRR1 complex structure shown in *Figure 5D*.

**Figure supplement 1.** ancHK-ancRR1 AlphaFold structure confidence.

these results indicate that changes in just two of the four proteins, ancHK2 and ancRR1, which are notably not cognate partners, were required to insulate these systems (*Figure 5C*).

## Ancestral interaction was not optimized for rapid phosphotransfer

Although our results are sufficient to explain how ancHK2 developed paralog specificity, it remained unclear how ancHK1 developed its specificity after duplication. While ancHK1 exhibits more cross-talk than ancHK2, it does exhibit a slight kinetic preference for ancRR1 over ancRR2 (*Figure 3A*). Because there were no differences in the six strongly covarying residues between ancHK and ancHK1 or between ancRR and ancRR2 (*Figure 4A*), we hypothesized that mutations in ancRR1 must have been responsible for this specificity change. Indeed, when we introduced the substitution R11E into ancRR, we observed slower transfer from ancHK2, as already noted (*Figure 4F, I*), as well as faster transfer from both ancHK1 and ancHK (*Figure 5A–B*). This result suggests that a single mutation in ancRR was sufficient to both improve the ancestral interaction and help prevent crosstalk with the new paralog ancHK2 (*Figure 5C*). This finding further suggests that the ancestral ancHK-ancRR

interaction was not optimized for the most rapid possible phosphotransfer, and that ancHK1 evolved a preference for ancRR1 by simply improving the ancHK1-ancRR1 interaction such that this transfer outcompetes crosstalk to ancRR2.

To better understand why the R11E mutation in ancRR might improve phosphotransfer from ancHK, we used AlphaFold2 (*Jumper et al., 2021*) to predict the structure of the ancHK-ancRR and ancHK-ancRR1 complexes (*Figure 5D*). These structures suggested that substituting an arginine at position 11 in ancRR with a glutamate enables ancRR1 to form a salt bridge with R27 in ancHK and ancHK1. The emergence of this salt bridge may explain why the R11E substitution improves the interaction of ancRR1 with ancHK and ancHK1.

Although we observed, and can largely account for, paralog specificity in the ancestors that arise shortly after the duplication event, the extant proteins in *C. crescentus* exhibit more paralog specificity (*Figure 1D*), suggesting that subsequent mutations further insulated these paralogous protein interfaces. In particular, we observed only a weak preference of ancHK1 for ancRR1 relative to ancRR2 (*Figure 3A*), while *C. crescentus* EnvZ1 has a much stronger preference for its cognate partner

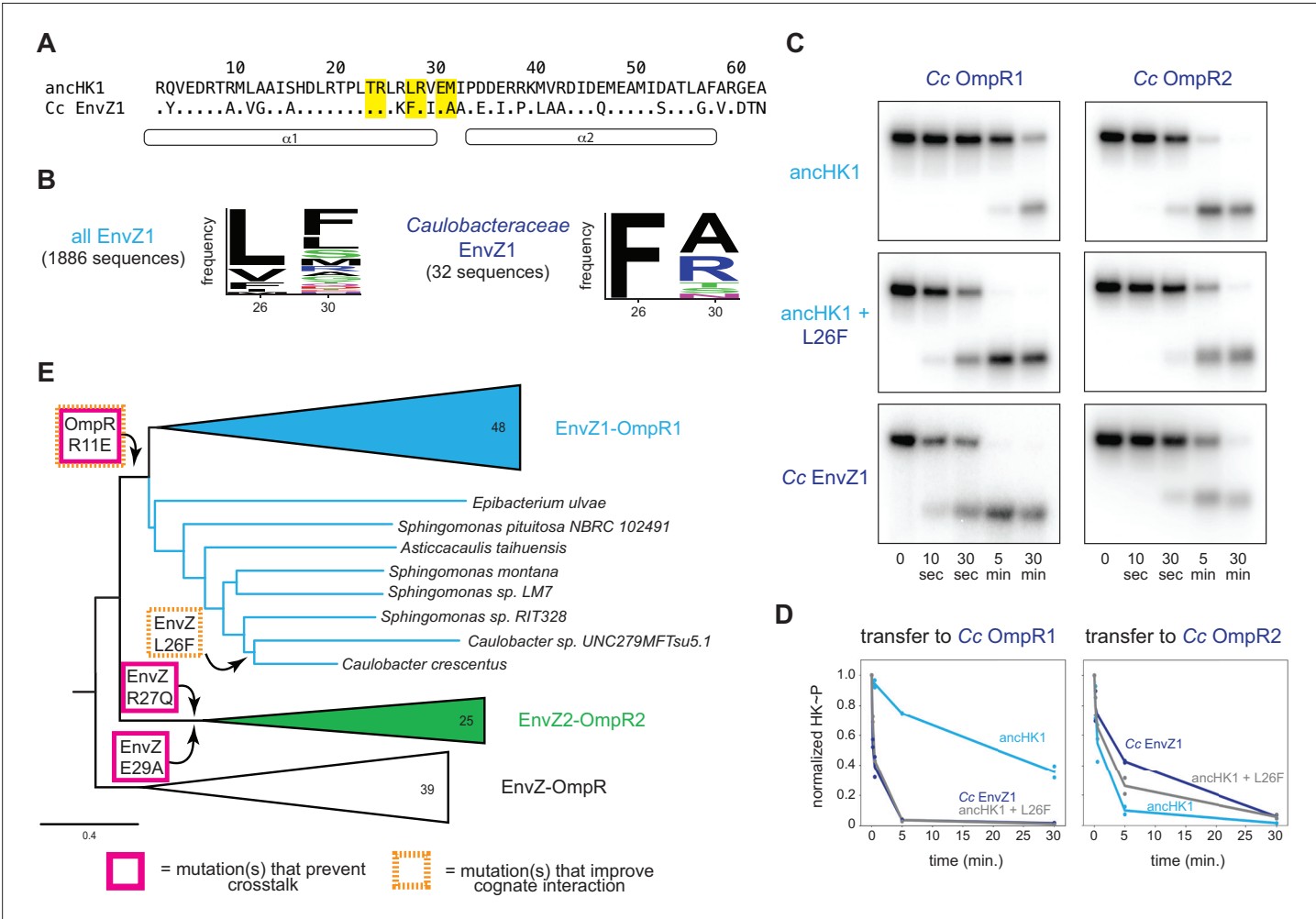

**Figure 6.** Tracing the mutations that produced extant, insulated EnvZ-OmpR paralogs. (**A**) Sequences of DHp domain for ancHK1 and *C. crescentus* EnvZ1 shown as in *Figure 4A*. (**B**) Sequence logos for HK positions 26 and 30 in 1886 identified EnvZ1 paralogs and 32 EnvZ1 paralogs from *Caulobacteraceae* species, with height indicating frequency of each amino acid. (**C**) Phosphotransfer from ancHK1, ancHK1 + L26F, and *C. crescentus* EnvZ1 to *C. crescentus* OmpR1 (left) and *C. crescentus* OmpR2 (right). (**D**) Quantification of normalized phosphorylated HK from (**C**). (**E**) Expanded, simplified phylogeny of α-proteobacterial EnvZ-OmpR paralogs showing origins of key historical mutations leading to *C. crescentus* and whether they affect crosstalk (pink) and/or cognate interaction (orange).

The online version of this article includes the following source data for figure 6:

**Source data 1.** Quantified phosphotransfer values for *Figure 6D*.

(*Figure 1D*). To identify residues in *C. crescentus* EnvZ1 that may have provided additional specificity, we compared the sequences of ancHK1 and *C. crescentus* EnvZ1 and identified two mutations in the strongly covarying residues (positions 26 and 30) that differ (*Figure 6A*). When we looked at the identity of these residues in close relatives of *C. crescentus*, the *Caulobacteraceae*, we find that one of these positions – position 26 – is highly conserved in this clade, with a phenylalanine in the *Caulobacteraceae* compared to a leucine in the ancHK1 ancestor (*Figure 6B*).

When we introduced this substitution into ancHK1, we observed faster transfer to *C. crescentus* OmpR1 (*Figure 6C*), recapitulating the behavior observed for *C. crescentus* EnvZ1 (*Figure 6D*). This mutation had a smaller effect on transfer to *C. crescentus* OmpR2 (*Figure 6C, D*), although it also slowed transfer to this non-cognate partner. This finding suggests that a leucine to phenylalanine substitution in the lineage leading from ancHK1 to EnvZ1 provided further insulation of the two paralogous systems (*Figure 6E*). This observation also supports the model that insulation of the two paralogous systems was accomplished primarily by improving the cognate ancHK1-ancRR1 interaction and breaking the non-cognate ancHK2-ancRR1 interaction (*Figure 5C*).

## Discussion

### Ancestral sequence reconstruction of protein-protein interactions

Ancestral sequence reconstruction has been used to interrogate the evolutionary history of a variety of protein functions, including DNA binding (*Anderson et al., 2015*; *McKeown et al., 2014*), small molecule binding (*Bridgham et al., 2009*; *Bridgham et al., 2006*), oligomerization (*Hochberg et al., 2020*; *Pillai et al., 2020*), and enzymatic activity (*Boucher et al., 2014*; *Howard et al., 2014*). But little has been done to investigate how gene duplication events impact protein-protein interactions. Here, we showed that this technique can be used to simultaneously reconstruct two interacting proteins, and we used these ancestors along with mutational intermediates to determine the evolutionary trajectory that allowed for the generation of new, insulated protein interaction interfaces.

In the case where a two-component signaling system, usually encoded as a bicistronic operon, is duplicated, two paralogous protein interaction interfaces are generated. If these two interaction interfaces are to ultimately support two separate signaling pathways, they must acquire mutations that somehow prevent detrimental crosstalk and ensure interaction specificity. For all two-component systems, and indeed for virtually all protein-protein interactions, the mutational trajectories responsible for establishing such specificity in paralogs post-duplication are unknown. Generally speaking, specificity can be generated via either (i) mutations in one of the paralogous systems such that the interacting proteins maintain their interaction while eliminating interaction with the paralogous system or (ii) mutations in proteins from both systems to generate specificity of both paralogous interfaces. In the case of the EnvZ/OmpR duplication in α-protebacteria, our results support the latter model that mutations in both paralogous systems were required to insulate these interfaces. Somewhat surprisingly, however, we found that changes in just two proteins (ancHK2 and ancRR1) were sufficient to establish specificity, rather than changes to all four proteins. This finding suggests that both protein interaction interfaces were impacted by the duplication event, and that a new two-component signaling system was not generated entirely via neofunctionalization of one system. Instead, proteins from both systems had to change to generate signaling specificity. It remains unclear if this mode of diversification represents the norm for duplicated two-component signaling systems, or duplicated interacting proteins more generally. Further characterization of ancestral interacting proteins will be required to determine whether this mechanism of insulation is commonplace.

### A small set of substitutions was sufficient to generate insulated protein interfaces

Previous work has shown that a small number of historical substitutions can have significant consequences for the specificity of protein binding to DNA (*McKeown et al., 2014*), protein binding to small molecules (*Bridgham et al., 2006*), and protein multimerization (*Finnigan et al., 2012*; *Pillai et al., 2020*). For two-component signaling systems, substitutions in just a few key residues of the histidine kinase or response regulator can significantly alter the specificity of their interactions and generate new synthetic insulated protein-protein interactions (*Capra et al., 2010*; *McClune et al., 2019*; *Skerker et al., 2008*). However, it remained unclear if such small sets of mutations were

sufficient to establish specificity upon a gene duplication event, and whether the ancestral mutations that drove paralog insulation involve the same residues involved in rewiring the specificities of extant two-component systems. We found that, for the EnvZ-OmpR paralogs in α-proteobacteria, just three key ancestral substitutions (R27Q and E29A in ancHK2, and R11E in ancRR1) were indeed sufficient to establish specificity upon duplication, with one additional subsequent mutation (L26F in ancHK1) sufficient to establish specificity that resembles the specificity observed in the extant *C. crescentus* systems. These findings indicate that new, insulated two-component signaling pathways can readily evolve via gene duplication and the subsequent accumulation of just a few key substitutions.

The insulation of the paralogous ancHK1-ancRR1 and ancHK2-ancRR2 interfaces was primarily accomplished by weakening the interaction between ancHK2 and ancRR1, as well as strengthening the interaction between ancHK1 and ancRR1. This evolutionary path relied on the fact that the ancestral interaction, ancHK-ancRR, was not fully optimized for rapid phosphotransfer. Although this interaction was likely fully functional prior to the duplication event, the ability of the R11E substitution in ancHK to increase the rate of transfer from ancHK (and ancHK1) allowed the diversification of these paralogous interfaces by strengthening one of the two cognate interactions. We speculate that other two-component signaling systems and other protein-protein interactions that are similarly non-optimal may be particularly well-suited to duplication and divergence. It is worth noting that we cannot rule out the possibility that the lack of optimality we see in the ancestral sequences may be due to errors in the ancestral sequence reconstruction. However, we believe this is unlikely as we have identified two independent mutations (R11E in ancRR and L26F in ancHK1) that improve the ancHK1-ancRR1 interface.

## How novel protein-protein interactions evolve

Both selection and neutral drift are important in the generation of evolutionary novelty. For two-component signaling systems, it remains an open question whether the changes that result in insulation of these systems generally accumulate slowly over evolutionary time through drift, or whether strong selection drives insulation of these systems. In the case of α-proteobacterial EnvZ-OmpR systems, we found that after a gene duplication event there was a rapid change in just a few key residues that dictate protein interaction specificity. Subsequent to this burst of changes, there were many mutations that accumulated in these paralogous proteins, but these mutations do not seem to have made major contributions to the insulation of these systems. This sequence of events suggests that strong selection against crosstalk occurs immediately post-duplication followed by long periods of relative stasis in the key specificity-determining residues and neutral accumulation of changes elsewhere in these proteins.

We have determined the likely evolutionary trajectory that resulted in the diversification of the two paralogous EnvZ-OmpR paralogous systems in α-proteobacteria. In this case, a small set of mutations in two non-cognate proteins was largely sufficient to establish insulation of the two pathways. Whether similar trajectories have been followed to establish other paralogous two-component signaling pathways remains an open question, but it seems unlikely that a single model will account for the mechanism of divergence of these systems in general. Our work also focused entirely on the two systems that were produced by a duplication event. However, prior work has indicated that the avoidance of crosstalk with other, existing paralogs following a duplication event can also select for changes in specificity residues (*Capra et al., 2012*). Further studies of how paralogs emerge, using similar ancestral reconstruction methods as used here, promises to shed more light on the general principles and mechanisms by which two-component signaling pathways, and other protein-protein interactions found throughout biology, evolve.

## Methods

### Ancestral protein reconstructions

EnvZ and OmpR homologs from the ProGenomes database (*Mende et al., 2020*) were identified using HMMER (*Eddy, 2011*). Cognate histidine kinase and response regulator pairs were matched based on genome proximity, with only adjacent genes matched. Clusters of HK-RR genomic sequences, where it was difficult to identify cognate pairs, were removed from the analysis. Protein sequences were then merged into a concatenated sequence for each matched pair. A subset of representative merged

sequences (200 total) were aligned with MUSCLE (*Edgar, 2004*), the N-terminal sensory domain from EnvZ was removed, and sequences were re-aligned with MUSCLE. The best fit evolutionary model was selected using ModelTest (*Darriba et al., 2020*) and the Akaike Information Criterion (LG +gamma) and a maximum likelihood phylogeny was inferred using PhyML 3.3.3 (*Guindon et al., 2010*). Node support was evaluated using the approximate likelihood ratio test statistic (in PhyML); tree was rooted on Actinobacteria EnvZ-OmpR homologs MprBA. Ancestral sequences were then reconstructed using the codeml package in PAML 4.8 (*Yang, 2007*) using the maximum likelihood phylogeny (full DNA sequences for all reconstructed ancestors in *Supplementary file 1*). This process was repeated with the HK-only and RR-only alignments to determine individual phylogenies (*Figure 2—figure supplement 2*) and ancestral sequences based on these phylogenies (*Figure 2—figure supplements 4 and 5*). To account for uncertainty in the reconstructions, ambiguously reconstructed sites were identified as those at which multiple residues had posterior probabilities >0.2 (*Eick et al., 2012*). For each ancestral protein, an alternative ancestor was generated by incorporating the second highest likelihood residue at all ambiguous sites (*Figure 3—figure supplement 2*, full sequences in *Supplementary file 1*).

## Identification of paralog-specific and species-specific residues

To identify paralog-specific residues in a larger set of EnvZ-OmpR sequences, a merged concatenated HMMER-aligned sequence was generated for all matched protein pairs identified using HMMER and genome proximity, as described above (~11,000 total sequences). A phylogenetic tree was constructed using FastTree (*Price et al., 2009*) and EnvZ1-OmpR1 and EnvZ2-OmpR2 paralogs were classified based on clade identity. To identify *Caulobacteraceae*-specific EnvZ1 residues in this same set of sequences, EnvZ1 paralogs were identified that were members of the *Caulobacteraceae* based on species classification from the ProGenomes database (*Mende et al., 2017*).

## Species tree

To determine the distribution of EnvZ-OmpR paralogs, a proteobacterial species tree was generated based on a concatenated alignment of 27 ribosomal protein genes (*rpsD, rplD, rpsC, rplF, rpsK, rplA, rplI, rpsG, rplP, rplX, rpsH, rplJ, rplK, rplT, rplM, rpsI, rplB, rplV, rpsE, rplO, rpsA, rpsB, rpmE2, rpsF, rpsT, rplU, rplQ*) (*Hug et al., 2016*). HMMER was used to identify and align orthologs of these genes from the ProGenomes Database. The concatenated alignment was manually trimmed to remove positions represented in <50% of sequences and positions with <25% conservation, and a tree was generated using FastTree, and rooted on Cyanobacteria (*Supplementary file 2*). EnvZ-OmpR distribution was determined by identifying matched EnvZ-OmpR pairs from the protein phylogeny described above. Visual inspection of a species tree generated in *Parks et al., 2018* suggests that use of this newer tree would likely not change our results.

## Protein expression and purification

Expression and purification of EnvZ and OmpR and ancestral proteins was carried out as previously described (*Skerker et al., 2005*). Briefly, the cytoplasmic domains (DHp and CA) of EnvZ were purified fused to an N-terminal MBP-His$_6$ tag; full-length OmpR was purified fused to an N-terminal Trx-His$_6$ domain. Both proteins were expressed in BL21(DE3) cells and purified on a Ni$^{2+}$-NTA column.

## Phosphotransfer assays

Phosphotransfer experiments were carried out as previously described (*Skerker et al., 2005*). Briefly, a given histidine kinase was first autophosphorylated with [γ-$^{32}$P]-ATP (Perkin Elmer) for 90 min at 30 °C to drive autophosphorylation and then mixed at a 1:4 molar ratio with a response regulator (1 μM EnvZ, 4 μM OmpR). Reactions were incubated at 30 °C and stopped at relevant timepoints by adding 4 x Laemmli buffer with 8% 2-mercaptoethanol. Products were separated by SDS-PAGE (BioRad Any kD Mini-PROTEAN TGX Gel), exposed to a phosphor screen, and quantified with a Typhoon scanner (GE Healthcare) at 50 μm resolution. A representative image of two independent experiments is shown in figures. Images were quantified using ImageQuant, with rolling ball background subtraction (radius=200), and normalized to t=0 lane for each HK-RR pair. To quantify kinetic preferences, initial rates of phosphotransfer were determined. (*Skerker et al., 2008*). Initial rates were determined by measuring the rate of loss of phosphorylated kinase between 0 and 30 s for cognate

substrates, and between 0 and 5 min for all other HK-RR pairs. For in vitro competition experiments, for increased visibility of RR bands, autophosphorylated *C. crescentus* EnvZ1 was mixed with *C. crescentus* OmpR1 and OmpR2 at a 2:1 molar ratio (8 µM EnvZ, 4 µM OmpR1, 4 µM OmpR2). After exposure to a phosphor screen, the gel was stained with Coomassie brilliant blue to distinguish response regulators by size.

### Protein structure prediction

The predicted structure of the ancHK-ancRR and ancHK-ancRR1 complexes was generated using AlphaFold2 (*Jumper et al., 2021*), modeling the histidine kinase as a homodimer and the response regulator as a monomer (*Figure 5—source data 2*). Default parameters were used (MSA method: mmseqs2, pair mode: unpaired, number of models: 5, max recycles: 3).

## Acknowledgements

We thank C McClune, I Frumkin, and D Ghose for valuable discussions and comments on the manuscript; B Wang for assistance with phosphotransfers; and J Thornton for advice on ancestral reconstructions. This work was supported by an NIH grant (1F32GM126765-01) to IN.

## Additional information

### Competing interests

Michael T Laub: Reviewing editor, eLife. The other author declares that no competing interests exist.

### Funding

| Funder | Grant reference number | Author |
| --- | --- | --- |
| National Institute of General Medical Sciences | 1F32GM126765 | Isabel Nocedal |
| Howard Hughes Medical Institute | | Michael T Laub |

The funders had no role in study design, data collection and interpretation, or the decision to submit the work for publication.

### Author contributions

Isabel Nocedal, Conceptualization, Data curation, Formal analysis, Funding acquisition, Investigation, Methodology, Visualization, Writing – original draft, Writing – review and editing; Michael T Laub, Conceptualization, Funding acquisition, Project administration, Supervision, Writing – original draft, Writing – review and editing

### Author ORCIDs

Isabel Nocedal ⓘ http://orcid.org/0000-0002-4706-1113
Michael T Laub ⓘ http://orcid.org/0000-0002-8288-7607

### Decision letter and Author response

Decision letter https://doi.org/10.7554/eLife.77346.sa1
Author response https://doi.org/10.7554/eLife.77346.sa2

## Additional files

### Supplementary files

• Supplementary file 1. Excel spreadsheet containing relevant strains, primers, and protein sequences.

• Supplementary file 2. Newick file of proteobacteria species tree (shown in *Figure 1C*). Species numbers from ProGenomes database (http://progenomes.embl.de/index.cgi).

• Supplementary file 3. Newick file of EnvZ/OmpR merged phylogeny used for ancestral

reconstructions (shown in *Figure 2—figure supplement 1*). Protein numbers from ProGenomes database.

• Transparent reporting form

## Data availability

All data generated or analysed during this study are included in the manuscript and supporting files. All gels are provided as Source Data with a master file indicating which individual file contains each gel that appears in the paper. All data plotted in graphs are provided in Excel files.

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
