## [Editor Report]

It is thought that gene duplications are a major factor driving the emergence and expansion of paralogous proteins, but how redundant proteins evolve to acquire new interaction specificities is poorly understood. Here the authors focused on understanding how specificity evolves in signaling pathways involving bacterial two-component systems. They identify three residues involved in preventing cross-talk between non-cognate pairs of two pairs of paralogous proteins. Remarkably, they show that changes in these 3 residues were sufficient for the establishment of specificity and establishment of insulation of two signaling pathways following a duplication event.

---

## [Decision Letter]

**Decision letter after peer review:**

Thank you for submitting your article "Ancestral reconstruction of duplicated signaling proteins reveals the evolution of signaling specificity" for consideration by *eLife*. Your article has been reviewed by 4 peer reviewers, including Karina B Xavier as Reviewing Editor and Reviewer #1, and the evaluation has been overseen by Sara Sawyer as the Senior Editor.

The reviewers have discussed their reviews with one another, and the Reviewing Editor has drafted this to help you prepare a revised submission. In general, all four reviewers found the manuscript solid and the results exciting, and thus suitable for publication for *eLife*. However, there is still a major point that needs additional clarification. Below is a list of essential revisions to help improve the clarity of the work. Additionally, you can also find at the end of this letter the separate "recommendations for the authors" written by each of the reviewers which should help to improve the manuscript and to understand the rationale of each of the reviewers. We hope you will use this information to improve your manuscript, but you do not need to reply to each of the individual comments from the reviewers, you only need to reply the essential points summarized here.

Essential revisions:

1) Reviewer 3 raises concerns about the approach used in the present study for the ancestral sequence reconstruction and proposes an alternative approach. We recommend that the authors consider/discuss the approach proposed (see first weakness raised by reviewer #3). It is unclear if this approach will change the results significantly, and if so, if additional experiments would be needed. It is important that the authors clarify this point by responding to this letter, while preparing a revised submission. Depending on the results additional experiments might be needed. In the case that additional experiments are necessary, we invite you to respond with a proposed plan for the additional experiments required to revise the manuscript.

*Reviewer #1 (Recommendations for the authors):*

I found the paper clear and technically sound. I have no major recommendations.

*Reviewer #2 (Recommendations for the authors):*

1) The authors' propose (based on their data for EnvZ-OmpR) that "strong selection against crosstalk occurs immediately post-duplication followed by long periods of relative stasis." I agree with this interpretation, but feel it would be bolstered by more biological context. It would be helpful to include a description of the biological role of EnvZ-OmpR, and some discussion regarding the possible fitness costs of cross-talk (or conversely, fitness benefits conferred by evolving a new paralogous pair).

2) The authors state that the extant EnvZ-OmpR pairs (EnvZ1-OmpR1 and EnvZ2-OmpR2) are only 50% identical to each other and that the ancestral sequences are only 50% identical to the extant sequences. However, I would appreciate more discussion of sequence identity early in the results section entitled 'Ancestral protein reconstruction reveals early acquisition of paralog specificity'. In particular, how identical are ancHK1-ancRR1 to ancHK2-ancRR2, and how identical are these two pairs to ancHK-ancRR?

*Reviewer #3 (Recommendations for the authors):*

I have a number of suggestions on the presentation:

1. I found myself flipping back and forth between Fig 2, Fig 3, Fig 4, Fig S5 a lot. It might make the overall patterns clearer to summarize the various specificities in a large HK x RR graphic. Maybe have ancHK, ancHK1, ancHK2, ancHK+R27Q/E29A on the left and ancRR, ancRR1, ancRR2, ancRR+R11E on the top? You could either show results with gels, or maybe with a heat map where intensity represents activity?

2. On Fig 3F and S4 the fold-preference color scale threw me off for a bit. I think this is because the mutation/arrow side of the graphic encodes HK/RR vertically and fold preference horizontally; the color bar encodes fold preference vertically. Maybe put color bar horizontal on the bottom?

3. You mentioned alternate reconstructions on p. 9, but didn't introduce those alternate reconstructions until p. 11. Maybe add a sentence about alternate reconstruction to the initial description of ASR?

4. The supports were illegible in Fig S2. In the main text, the "*" notation works for your aLRT. In the supplement, it would be helpful to see the raw values, at least for the main ancestors.

5. Related to #4, it would be helpful to report the mean posterior probabilities for the ancestors in the main text, rather than making readers dig through Fig S4.

*Reviewer #4 (Recommendations for the authors):*

1) The authors used a rather outdated approach to building the bacterial species tree (Hug et al., 2016). A more robust, standardized bacterial taxonomy is available (DH Park et al., 2018 Nat Biotech), including download options (https://gtdb.ecogenomic.org/). This should not affect the overall results, given the fact that relationships on the class level (alphaproteobacteria) revealed by the two methods should be similar, but it might be worth checking.

2) Fig. 1C suggests that both paralogous systems are the subject to gene loss in a somewhat equal manner. However, in Fig. 2A (and Suppl. Fig. 2), the number of sequences in EnvZ2-OmpR2 cluster is twice smaller than in EnvZ1-OmpR1 cluster. How representative Cluster 2 is compared to Cluster 1? Was there a sporadic loss throughout alphaproteobacteria or is the system missing from a specific, rather large clade? The latter potentially might skew the comparison.

3) Suppl. Fig. 2: ancestors of EnvZ2-OmpR2 should be labeled as "ancHK2-ancRR2" (in green).

---

## [Author Response]

Essential revisions:1) Reviewer 3 raises concerns about the approach used in the present study for the ancestral sequence reconstruction and proposes an alternative approach. We recommend that the authors consider/discuss the approach proposed (see first weakness raised by reviewer #3). It is unclear if this approach will change the results significantly, and if so, if additional experiments would be needed. It is important that the authors clarify this point by responding to this letter, while preparing a revised submission. Depending on the results additional experiments might be needed. In the case that additional experiments are necessary, we invite you to respond with a proposed plan for the additional experiments required to revise the manuscript.

We appreciate this concern and have added additional analysis to address it. To ensure the concatenated phylogenies do not contain artifacts, we have built independent phylogenies based on HK-only and RR-only alignments. We find that the trees are largely concordant, with no changes in the membership of the EnvZ1/OmpR1 and EnvZ2/OmpR2 clades studied here, and only subtle changes in the placement of some of the outgrouping sequences. This is now shown in a new supplemental figure (Figure 2 —figure supplement 2) and referenced in the Results (first paragraph of “Ancestral protein reconstruction reveals early acquisition of paralog specificity”).

Furthermore, we have computationally reconstructed ancestors based on these HK- and RR-only phylogenies and find them to be highly similar to those based on our concatenated alignment, with no changes in the key residues identified here as important in diversification. Importantly, many of the positions that vary between the matched and individual phylogenies have already been mutated in our “alt-all” alternative ancestors. Here are the exact numbers for each of the major ancestors in the paper:

ancHK: differs at 11 positions, 8 already tested in alt-all ancHK1: differs at 23 positions, 17 already tested in alt-all ancHK2: differs at 12 positions, 11 already tested in alt-all ancRR: differs at 26 positions, 16 already tested in alt-all ancRR1: differs at 28 positions, 14 already tested in alt-all ancRR2: differs at 17 positions, 13 already tested in alt-all

Given that our “alt-all” ancestors incorporated between 29 and 43 point mutations and still behaved similarly, we believe it is unlikely that the relatively small number of outstanding changes are likely to produce significantly different specificity. We have added two additional supplemental figures (Figure 2 —figure supplements 4 and 5) to show the alignments for these HK-only and RR-only ancestors compared to our matched ancestors and the alt-all ancestors and which positions differ between them. We have also added a sentence to the Results section discussing the similarity between the sequences produced from these different methods of analysis (end of paragraph 1 of “Ancestral protein reconstruction reveals early acquisition of paralog specificity”).

Reviewer #2 (Recommendations for the authors):1) The authors' propose (based on their data for EnvZ-OmpR) that "strong selection against crosstalk occurs immediately post-duplication followed by long periods of relative stasis." I agree with this interpretation, but feel it would be bolstered by more biological context. It would be helpful to include a description of the biological role of EnvZ-OmpR, and some discussion regarding the possible fitness costs of cross-talk (or conversely, fitness benefits conferred by evolving a new paralogous pair).

Added discussion of cross-talk to introduction.

2) The authors state that the extant EnvZ-OmpR pairs (EnvZ1-OmpR1 and EnvZ2-OmpR2) are only 50% identical to each other and that the ancestral sequences are only 50% identical to the extant sequences. However, I would appreciate more discussion of sequence identity early in the results section entitled 'Ancestral protein reconstruction reveals early acquisition of paralog specificity'. In particular, how identical are ancHK1-ancRR1 to ancHK2-ancRR2, and how identical are these two pairs to ancHK-ancRR?

Added this to the results section.

Reviewer #3 (Recommendations for the authors):I have a number of suggestions on the presentation:2. On Fig 3F and S4 the fold-preference color scale threw me off for a bit. I think this is because the mutation/arrow side of the graphic encodes HK/RR vertically and fold preference horizontally; the color bar encodes fold preference vertically. Maybe put color bar horizontal on the bottom?

We appreciate this suggestion and have changed this in Figure 3 and S4 (now Figure 3 – Figure Supplement 2).

4. The supports were illegible in Fig S2. In the main text, the "*" notation works for your aLRT. In the supplement, it would be helpful to see the raw values, at least for the main ancestors.

The raw values have been added to Fig S2 (now Figure 2 – Figure Supplement 1).

5. Related to #4, it would be helpful to report the mean posterior probabilities for the ancestors in the main text, rather than making readers dig through Fig S4.

This has been added to the results section.

Reviewer #4 (Recommendations for the authors):2) Fig. 1C suggests that both paralogous systems are the subject to gene loss in a somewhat equal manner. However, in Fig. 2A (and Suppl. Fig. 2), the number of sequences in EnvZ2-OmpR2 cluster is twice smaller than in EnvZ1-OmpR1 cluster. How representative Cluster 2 is compared to Cluster 1? Was there a sporadic loss throughout alphaproteobacteria or is the system missing from a specific, rather large clade? The latter potentially might skew the comparison.

There was sporadic loss throughout the alpha-proteobacteria, with EnvZ2/OmpR2 lost more frequently than EnvZ1/OmpR1. The species shown in Figure 1C do not correspond 1:1 with the sequences used in Figure 2C, hence the discrepancy. However, highly similar sequences were removed from the phylogenetic analysis to ensure that the analysis was not skewed by oversampling of particular clades.

3) Suppl. Fig. 2: ancestors of EnvZ2-OmpR2 should be labeled as "ancHK2-ancRR2" (in green).

This has been corrected.